# Contextualising the Effectiveness of an Employee Assistance Program Intervention on Psychological Health: The Role of Corporate Climate

**DOI:** 10.3390/ijerph19095067

**Published:** 2022-04-21

**Authors:** Sophia Bouzikos, Ali Afsharian, Maureen Dollard, Oliver Brecht

**Affiliations:** 1PSC Global Observatory, Centre for Workplace Excellence, Justice and Society, University of South Australia, Adelaide, SA 5001, Australia; sbouzikos@gmail.com (S.B.); maureen.dollard@unisa.edu.au (M.D.); 2Employee Assistance Professional Association of Australia, Willoughby, NSW 2068, Australia; oliver@veretis.com.au

**Keywords:** psychosocial safety climate, employee assistance programs, client satisfaction, psychological distress, work stress

## Abstract

Organisations often engage Employee Assistance Programs (EAPs) to assist employees experiencing psychological distress, yet EAPs primarily focus on individual remedies rather than addressing the context of the problem (e.g., the corporate climate) which may render them limited in effectiveness. We investigated the effectiveness of EAPs and the role of organisation psychosocial safety climate (PSC) (the corporate climate for worker psychological health and safety) and client satisfaction in reducing client psychological distress. Client participants (Time 1, *n* = 100, Time 2, *n* = 28, Matched *n* = 25) from Australia and New Zealand entering an EAP took part in two online surveys, pre- and post-EAP, around five weeks apart. Multilevel analysis showed a significant reduction in psychological distress due to the EAP (individual effect) but particularly at high levels of PSC (organisational effect). Thus, EAPs could engender a more significant impact by also assisting organisations to improve their PSC (i.e., through implementation of policies, practices and procedures for worker psychological health and safety), in combination with individual interventions.

## 1. Introduction

At least in the developed world, many corporations offer Employee Assistance Programs (EAPs) to employees in need of psychological help [1]. Workers experiencing psychological health problems may access EAPs for professional assistance with personal or work-related issues that may be negatively impacting their work performance or employment. EAPs provide work-based interventions (such as individual counselling) intended to improve employees’ emotional and psychological well-being and support their immediate family members [2]. The problem is that the presenting psychological health issue may have its basis in work and organisational factors such that individual-level approaches may have limited value.

The psychosocial safety climate (PSC) refers to the organisational climate for employee psychological health and safety and encompasses the way management value and prioritise worker psychological health. PSC theory suggests that the corporate climate for psychological health plays a significant role in determining whether a workplace is stressful. Occupational stress is one of the main reasons for mental health injury and may result from pressures within the workplace [3]. Without giving appropriate attention to the root cause of distress, such as the corporate climate, potential gains in mental health due to EAP consultations may be limited, particularly when employees return to a stressful context. This may have costly consequences for the individual and the employer. Poor psychological health and wellbeing is the leading cause of sickness absence for Australian workers [4].

According to the International Employee Assistance Professionals Association, the aim of EAPs is to “serve organisations and their employees in multiple ways ranging from a consultation at the strategic level about issues with organisation-wide implications to individual assistance to employees and family members experiencing personal difficulties” [5] p. 2. However, little evidence-based information and few recommendations inform how and what EAPs and workplaces should do to integrate and incorporate the use of EAPs for more effectiveness [6].

Our study aimed to contextualise the effectiveness of EAPs on the psychological health and wellbeing of employees by considering the role of corporate climate and client satisfaction as potential moderators. PSC theory emphasises how worker psychological health is valued in the workplace and, if considered in conjunction with EAP services, could provide insights into the effectiveness of the EAP. Furthermore, less satisfied EAP clients may also benefit less from EAP treatment, implying a moderation due to EAP quality. Understanding EAP effectiveness in context may assist in identifying and integrating individual and organisational pathways to improve employee psychological health and wellbeing, and may provide desirable impetus to inform employers, organisational policy and EAP providers.

### 1.1. Psychological Health and Wellbeing

Poor psychological health and wellbeing can have adverse personal and occupational effects dependent upon workplace stressors experienced by employees. Stressors may include work overload, bullying, relational conflicts, perceived lack of control, work/life balance difficulties and poor work environment [3,7]. A lack of work wellbeing influences work motivation, job satisfaction, product quality and productivity [8]. In Australia, approximately 7230 workers’ compensation claims are paid per year for work-related severe mental health conditions and most (93%) are caused by psychological distress. The calculated cost of work-related mental health conditions to the Australian economy per year is estimated to be around AU $476 million [4].

Investigating how to prevent and manage poor psychological health and wellbeing due to workplace phenomenon is exceptionally timely considering that many organisations are ill-equipped to support their employees. For example, of 379 organisational leaders involved in a workplace mental health study, approximately 25% reported difficulty with confidently and effectively supporting employees with poor emotional wellbeing [9,10]. Given that approximately 4 million working Australians (39%) score high levels of emotional exhaustion [11], and societal prevalence rates indicate one in five employed adults have mental health disorders [12], there is a clear gap in organisational capability to manage employee mental health needs.

In response to the above concerns, health advocates and policymakers have promoted the need for workplace initiatives, such as EAPs. However, despite their potential benefits, little is currently known about these programs, due to a lack of practical evaluation [6]. By addressing the psychological health status of the workforce and focusing more on prioritising workplace wellbeing strategies at the organisational level, employers and EAP providers may contribute to positive business growth and employee engagement, and encourage increased productivity and decreased absenteeism in organisations [13,14,15].

### 1.2. Employee Assistance Programs

To understand how EAPs could be improved, we need first to understand the role and purpose of EAPs and how they are used in organisations. EAPs are a type of workplace intervention initially created as a result of occupational drug and alcohol problems [16], but were then further designed to support staff, mitigate risk [17], address wider mental health issues and assist employees with personal and work-related problems that affect work performance. These include, but are not limited to, marital issues, financial stressors, family/relationship issues, emotional stress, work stressors, addictions, grief, trauma and serious illnesses [18,19]. EAPs provide services for individuals, families, management and organisations as a whole, and have increased in prominence in recent years [6,20].

EAP services can be offered within different organisations by a diverse array of professionals including chaplains, counsellors, social workers or psychologists [21,22]. EAPs include interventions and services such as workplace mediation, facilitation, counselling and supervisor mentoring, and may also help set organisational policies and procedures. Some services provide coaching to management and critical incident management support, which may involve debriefing, face-to-face or telephone counselling and trauma training [17,19]. The support and information provided to clients via assessment, education and referrals is intended to help employees cope with stress, and employers to identify distress and respond appropriately.

EAPs have been designed with the potential to improve employee psychological health and wellbeing [8], highlighting the need, benefit and value of rehabilitating employees rather than replacing them [23]. EAPs cost Australian employers $265 million per annum [24]. Therefore, the continued development and promotion of EAPs requires an ongoing evaluation of EAP efficacy, not only for improving workers’ psychological health and wellbeing but also because of the cost.

Several studies have examined the impact, benefit and effectiveness of EAPs. The overall performance of an EAP refers to “the effectiveness of EAPs in enhancing employee and organisational outcomes” [25]. In general, most existing research has used cost-benefit analyses to assess the ratio of costs incurred in conducting EAPs to the benefits after implementing them [12]. Lo Sasso et al. [26] found that providing treatment to depressed individuals had a “meaningful return on investment”. However, there are limitations with this evaluation approach due to the lack of pre- and post-test measures of mental wellbeing, leaving a substantial reliance on estimations of productivity gains and cost of the problem. Further, this narrow evaluative focus may support EAPs to become increasingly profit oriented, increasing the provision of cheap services with inadequately trained staff, and contributing to progressively poor-quality services [27].

In 2001, the British Association for Counselling and Psychotherapy published a report on workplace counselling, highlighting 16 studies supporting the effectiveness of workplace counselling on mental health with varied presenting problems relative to different organisational settings [28]. However, methodological limitations of the review such as the inclusion of low-quality studies and poor assessment methods indicate the findings may be significantly compromised. A further systematic review [29] supported workplace counselling as an intervention for reducing psychological problems. While consistent short-term benefits were reported, longitudinal follow-up data is necessary.

Similar to the current study design, a pre- and post-test study by Dickerson et al. [30] found that clients showed improvements and significant short-term benefits in emotional wellbeing after EAP intervention, when symptoms were compared from pre- and post-counselling [31]. Furthermore, employees using EAPs showed improvement in emotional wellbeing symptoms five months post interventions compared to a control group that did not use EAPs [8]. A study performed in the UK also confirmed the effectiveness of EAP counselling, where 70% of the sample showed recovery post-intervention [32]. In a large-scale longitudinal study, Attridge [12] found that EAP use was associated with increased productivity, engagement and life satisfaction, and reduced absenteeism and workplace distress. Previous studies support this finding and suggest improvements in physical and emotional health and relationships [8,33,34].

The evaluation of EAPs is crucial in developing interventions and ensuring the quality of the interventions. Authors have argued that the rapid growth of EAPs worldwide may have resulted from the need to account for and respond to increased consciousness of stress as a workplace phenomenon [35]. However, the popularity and proliferation of EAPs has not necessarily been matched with thoroughgoing research on their effectiveness in improving psychological health and wellbeing [28,36], leaving a noticeable gap in the present literature. Therefore, there is a pressing need for further informed and evidence-based initiatives. By addressing this gap in the research, employers and EAP providers may be better equipped to implement evidence-based practice.

**Hypothesis** **1.***The use of EAP individual sessions will significantly improve the psychological health and wellbeing of employees (indicated by a decrease in GHQ scores)*.

### 1.3. Psychosocial Safety Climate

As a facet of workplace climate, PSC refers to employees’ perception of the managerial policies, practices and procedures implemented in their organization to protect worker psychological health and safety and prevent psychological injury due to workplace psychosocial risks. Indeed, PSC protects employees’ psychological health and safety through four principles: management commitment, management priorities, organizational communications and participation [37].

In the evaluation of EAPs, there has been little consideration given to the workplace climate within which the EAP clients work. It is plausible that the workplace climate that clients return to could modify the effectiveness of EAPs. This research study will contribute to the developing body of EAP research by investigating the relationship between EAPs, their effectiveness and whether an organisation’s climate, such as PSC, could moderate EAP effects on employee outcomes.

Extensive evidence has linked employee psychological health and job performance to work design, work conditions and workplace culture [38]. An auxiliary observation by EAP practitioners implies that the success of EAPs is reliant on an organisation’s overall processes, practices and health [39]. The proposed goal of EAPs to create overall workplace wellness supplementary to an individual employee’s wellness is somewhat contradicted in that EAPs have a perpetually unfulfilled goal of workplace wellness without improving it through evaluation [40]. Bringing these threads together the current study will test the impact of organisational climate on EAP effectiveness by adopting the PSC theoretical framework and exploring PSC as a moderator of EAP effectiveness.

PSC concerns workplace policies, practices and procedures to protect the psychological health of employees. Research findings reported by the Australian Workplace Barometer (AWB) found PSC to be negatively related to all psychological distress measures [41], and research has shown PSC to be a reliable predictor of the psychological health of employees [42]. According to PSC theory [37], organisational policy and practices for worker psychological health directly influence psychosocial risk factors such as work pressure and poor job control, which in turn have a detrimental impact on worker psychological health [43,44]. Consider the scenario where an employee works in a poor PSC context, where there is no regard for worker psychological health, work pressure is high, and workloads are unpredictable. Following some helpful EAP sessions, the employee returns to work in a poor PSC context. PSC and EAPs likely interact with each other, affecting outcomes for individuals. This study will be the first to propose that the workplace climate could moderate the effectiveness of EAPs, and empirically testing this could add valuable content to the literature.

**Hypothesis** **2.***The effectiveness of EAPs as assessed by an increase in psychological health wellbeing (indicated by a decrease in GHQ scores) will be significantly stronger in workplaces with a high PSC (PSC-12). When PSC is low, the effect will not be as strong*.

### 1.4. Client Satisfaction

Client satisfaction is deemed a good benchmark to evaluate the success of EAPs [45]. Relatively high levels of client satisfaction with EAPs were reported by Shakespeare-Finch and Scully [46]. They signified that more than 50% of the workers benefited from EAP sessions (i.e., were satisfied). Compton and McManus [21] also indicated great satisfaction with EAPs and a close connection between EAPs and Human Resources Management strategy. Consistent findings from previous survey-based satisfaction studies show that when clients are satisfied overall with the EAP services provided, the specific counsellor and the EAP treatment, there are reported improvements in emotional health and quality of life [21,45,46,47]. To date there is little evaluation of the moderating role of client satisfaction with the EAP on their future psychological health. The reason satisfaction with the EAP service is likely to moderate the relationship between GHQ scores across time is that if the client has appraised the service as satisfactory they are likely to have found the counselling sessions to have provided helpful and supportive strategies for improving psychological wellbeing. For these clients it follows that psychological health is likely to improve following an EAP. By contrast if the service is rated as unsatisfactory or less satisfactory the client likely appraises the EAP strategies as unhelpful and as such their psychological health is less likely to improve. We expect that when clients are satisfied with the EAP service that a positive result would be more likely for them. In other words, if satisfaction is high then improvements should be greater than when they are low. We propose a second moderation hypothesis:

**Hypothesis** **3.***The effectiveness of EAPs as assessed by an increase in psychological health and wellbeing (indicated by a decrease in GHQ scores) will be significantly stronger when employees report high satisfaction with the service (measured with satisfaction scale), when measured post-EAP. When satisfaction is low the effect will not be as strong*.

## 2. Method

### 2.1. Participants

A convenience sample of organisations in Australia and New Zealand were invited via the Employee Assistance Professionals Association of Australasia (EAPAA) to voluntarily participate in this study. Through the EAPs, individual employees commencing an EAP were invited to participate and voluntarily opt into the study. The repeated measures sample comprised 25 participants: 9 from Australia and 16 from New Zealand. Eight participants were males and seventeen were females, aged 25–65 years. All participants were permanent workers on a fixed-term contract; 72% were employees, whilst 28% worked in managerial roles. Participants were from various industries, including government, health, manufacturing, construction, professional services, education and hospitality. The living location of participants varied, but the majority (92%) lived in the suburbs or urban locations. As an intervention check, we found that nine (36%) participated in one EAP session, ten (40%) in two sessions and six (24%) in three individual sessions for four to six weeks. A typical EAP session involved confidential counselling, usually short-term and solution focussed, to support worker wellbeing.

### 2.2. Design

We employed a longitudinal pre/post-test design collecting within subjects quantitative data using an online survey. The independent variable was time, which reflects the EAP intervention (individual sessions) delivered within an approximate one-month time frame. The dependent variable was employee psychological health and wellbeing (measured pre- and post-EAP). Moderating factors included PSC and client satisfaction.

### 2.3. Materials and Measures

Previous research has commonly assessed psychological health and wellbeing before and after counselling using a symptom checklist or mental health screening instrument [29]. We expand this to include demographic questions and self-report psychological distress, PSC and satisfaction.

#### 2.3.1. Demographic Measures

The demographic questionnaire included age, gender, employment status, living location, industry type and seniority.

#### 2.3.2. Psychological Distress

To assess the psychological health and wellbeing of participants, the General Health Questionnaire (GHQ-12) was administered. The GHQ-12 is widely used as a screening instrument for common mental disorders and is used as a general measure of mental wellbeing and psychological distress [48,49]. The GHQ-12 measures three domains; depression, anxiety and social dysfunction [50]. An example item is *“In the past 2 weeks, have you been able to concentrate on whatever you’re doing?”.* The items are measured on a 4-point Likert scale with scores from 0 to 4 with a typical response format, 0 = *better than usual* to 4 = *much less than usual* [51].

The Likert scoring method is used by assigning a value of 0, 1, 2 or 3 to each possible answer in the questionnaire, and then calculating the total score, where 36 is the highest possible total score [52]. The higher the total score on the GHQ-12, the greater the likelihood an individual will be diagnosed with a mental disorder [50]. We used the cut-off score of anything above 11 or 12 to identify those suffering from psychological distress [53]. The GHQ-12 in this study had excellent reliability (α = 0.92).

#### 2.3.3. Psychosocial Safety Climate

To assess the PSC of the workplaces of participants, the PSC-12 [54] was administered. The 12-item self-report scale assesses; (1) management support and commitment for stress prevention; (2) management priority to psychological health and safety versus productivity goals; (3) organisational communication; and (4) organisational participation and involvement [43,54]. An example item is “*Psychological well-being of staff is a priority for this organisation*”. Each sub-scale has three items measured using a 5-point Likert scale, ranging from 1 = *strongly disagree* to 5 = *strongly agree*. Scores from each subscale for all 12 items are combined to create a total score of 12 to 60. The PSC scores indicate low risk (≥41); medium risk, (>37–<41); high risk, (≤37–>26) and very high-risk, (≤26) PSC levels for depression [55]. The PSC-12 is associated with participant perceptions of psychosocial risk factors and health, and work-related outcomes [54]. Various samples have supported the scale as a reliable and valid measure across a range of occupations. The PSC-12 had excellent reliability (α = 0.97) in the current sample.

#### 2.3.4. Client Satisfaction

Previous literature that developed and used satisfaction scales to evaluate EAPs was reviewed to determine suitable items for integration into a questionnaire explicitly fashioned for use in the present study. Items from two existing satisfaction scales were contrasted to develop the current measure [47,56,57]. There was evidential overlap and consistency in both previous scales, highlighting the items’ relevance to the current study. The Client Satisfaction Questionnaire (CSQ) is an eight-item measure demonstrating good reliability and validity when previously used in various settings [58,59]. It had a high internal consistency and concurrent validity in mental health outpatient settings [60,61], with a Cronbach alpha of 0.92 [62]. The second measure, the EAP satisfaction test had a Cronbach alpha of 0.91, and a principal component analysis found apt psychometric rigour [56].

Based on the previous satisfaction measures, we selected four items that canvassed the main domains of client satisfaction, and that were deemed most important in the measure of EAP effectiveness. These domains included general satisfaction of the service delivered (EAP), whether the client would recommend the service, problem resolution and co-worker relationship dynamics. These items were also selected based on the strength of association with outcomes. An example item is *“Overall, I was satisfied with the service I received from the EAP”.* All items were measured on a five-point Likert scale, where 1 = *strongly disagree* and 5 = *strongly agree*. Total scores ranged from 4–20, where a high score indicated high satisfaction with the service. The satisfaction scale indicated good reliability and internal consistency (α = 0.86) with the current sample.

### 2.4. Procedure

The study design was developed in consultation with the EAPAA. The research proposal was presented to the service providers at their annual conference for feedback. Ethics approval was granted by the University of South Australia’s Human Research Ethics Committee (ethics protocol number: 202087). The participating organizations (EAP providers) were sourced via work email, phone and face to face contact. An information letter about the study was sent to organisations. A snowballing procedure was used, whereby providers received a survey link from an EAPAA representative and were asked to forward the link with the attached information sheet onto eligible clients within their EAP, specifically individuals wait-listed to commence an EAP. Participants accessed the study information and survey via a hyperlink available on mobile or computer devices directing them to the online survey. Participation was voluntary and consent was obtained if participants followed the prompts to complete the survey. Consenting participants were required to complete the questionnaire in one sitting prior to the EAP, and again in one sitting after a completed EAP one month later. Participants from various organisations commenced with different EAPs at different times and so the survey links were made accessible to employees for two weeks pre- and post-EAP. Participants were permitted to opt out of the study at any time. The average duration to complete the survey was four minutes. Participants were given a unique and confidential identification number that was used to link their survey data across time.

### 2.5. Data Analysis

Data were collected from Australian and New Zealand participants at pre- (*n* = 128) and post- (*n* = 28) EAP completion. At Time 2, 25 cases were successfully matched with data from Time 1 using participant email addresses. Data were cleaned, coded, screened and analysed using the SPSS-V26 software. For the final sample of 25 cases, after screening for normality, the study variables were not significantly skewed or kurtotic (standardised values of skew and kurtosis <1.96). Outliers were screened using Z-scores, and all values were found to be ≤2, *p* < 0.05, hence no values were removed. Initial correlations and ANOVAs were run to assess covariation and relationships between demographic variables (i.e., age, gender, industry type), GHQ, PSC and satisfaction measures.

Regarding Cohen [63], the correlation coefficients (*r*) were investigated which revealed the strength, direction and significant relationships between variables. Since the data were nested (repeated measures within individuals), we used hierarchical linear modelling (HLM) software [64] to analyse the hypotheses [65,66]. We tested a 2-level null model, which showed most of the variance in GHQ was due to the occasion accounting for 99.90% of the variance whereas 0.10% was due to the person. Nevertheless, we proceeded with a mixed effects model, since Bliese et al. [67] argue that these models are still appropriate with nested data even when the ICC-1 values are small and non-significant. We assessed change in GHQ from Time 1 to Time 2. We coded time into a dummy variable with T1 as the reference category which was entered at Level 1. PSC Time 1 was entered at Level 2 as a between-person variable. In the analysis the intercept is the average GHQ at Time 1 prior to the intervention; Time refers to GHQ at the respective times. The interaction of the PSC × Time variable indicates differences in GHQ effects across time due to PSC Levels (H.2). We repeated these analyses for satisfaction with EAP as the moderator (H.3).

## 3. Results

### 3.1. Descriptive Results

First, we assessed whether dropping out biased the sample. We tested whether the demographics of the drop out versus the stayer sample varied at Time 1. A Chi-Square test of independence looked at the association between dropping out (pre-test, dropped = 0, stayed = 1) on the ordinal demographic measures. The results showed no significant association for gender (χ^2^ = 1.03, *df* = 2, *p* > 0.05), living location (χ^2^ = 1.85, *df* = 2, *p* > 0.05), seniority (χ^2^ = 3.29, *df* = 3, *p* > 0.05) and employment status (χ^2^ = 3.29, *df* = 3, *p* > 0.05). The only significant result was found regarding age (χ^2^ = 18.68, *df* = 6, *p* < 0.01) and industry (χ^2^ = 34.11, *df* = 16, *p* < 0.01). Overall, these results indicate bias in the sample due to missingness was minimal.

Table 1 shows the means, standard deviations and correlations of the main study variables and correlations between them. A moderate positive correlation was found between GHQ scores at Time 2 and the number of EAP sessions completed by each participant, *r* = 0.45, *n* = 25, *p* < 0.05. A moderate negative correlation was observed between PSC scores at Time 1 and GHQ scores at Time 2, *r* = −0.53, *p* < 0.01, whereas cross-sectional relationships between GHQ and PSC scores at Time 1 and Time 2, respectively, produced weak and negative correlations that were not significant (Time 1, *r* = −0.17 and Time 2, *r* = −0.14). According to GHQ cut-off scores for psychological distress, the sample at Time 1 and Time 2 had a mean score much greater than 12 indicating high risk for diagnosis of a psychological disorder [53].

### 3.2. Hypothesis Results

Hypothesis 1 proposed that GHQ scores would significantly decrease following an EAP intervention. A paired-samples t-test revealed that there was a significant difference in GHQ scores from Time 1 (*M* = 22.40, *SD* = 6.80) to Time 2 (*M* = 12.21, *SD* = 7.22); *t* (24) = 5.73, *p* < 0.001, Cohen’s *d* = 1.45. This is also confirmed with a main effect of Time as shown in Table 2 (B = −10.19, *SE* = 1.66, *t* = −6.13, *p* < 0.001). Therefore, Hypothesis 1 was supported.

Hypothesis 2 proposed that PSC scores would moderate the relationship between GHQ scores from Time 1 and Time 2. When PSC and Time were entered as main effects both were significant. PSC was significantly associated with GHQ, B = −0.21, *SE* = 0.07, *t* = −3.10, *p* = 0.005. As shown in Table 2, after entering the main effects of Time and PSC, the PSC × Time interaction was significant, B = −0.21, *SE* = 0.10, *t* = −2.17, *p* = 0.04. The plot (see Figure 1) shows that under conditions of high PSC, change in GHQ (a reduction indicating an improvement in health) was greater when compared to change in low PSC contexts. To compare those two regression slopes, an additional analysis was performed to test simple intercepts and slopes at conditional values of PSC. The results indicated that both slopes were significant in high and low PSC situations. However, in high PSC (B = −12.87, *SE* = 1.91, *t* = −6.74, *p* < 0.001), a stronger significant slope was found compared with low PSC (B = −7.51, *SE* = 2.21, *t* = −3.40, *p* < 0.001). The initial significant interaction test implies that the slopes are significantly different.

Hypothesis 3 predicted that satisfaction with the EAP would moderate the relationship between GHQ scores across time. As shown in Table 3 there was no main or interactive effect of satisfaction with EAP on GHQ. Hypothesis 3 was not supported.

## 4. Discussion

This study analysed the effectiveness of EAPs on employee psychological health and contextualised potential effects by exploring the role of corporate climate and client satisfaction with the service. Our first hypothesis predicted that employee psychological distress would decrease after EAP treatment. The repeated measures pre- and post-test data revealed a significant reduction in psychological distress following an EAP intervention (ranging from one to three EAP sessions). This finding is also consistent with previous studies highlighting significant benefits after EAP sessions [28]. Prior studies have noted the vital role EAPs can have on clients’ improvements and significant short-term benefits [30], increased productivity, engagement and life satisfaction, and reduced absenteeism and workplace distress [8,12,33,34,35,36]. Next, we explored potential moderators of this effect over time. Very little was found in the literature on the EAP clients and their workplace climate. However, our findings support several relevant studies showing the association between workplace design, conditions and workplace culture with employees’ psychological health and performance [38] and employees’ health conditions concerning their organizations’ policies [39]. Our prediction that PSC would have a moderating effect on changes in employee psychological distress was supported. A decrease in employees’ psychological distress following an EAP was found to be moderated by the corporate climate (PSC). At high levels of PSC, the decrease in distress was stronger. Prior to testing the interaction effect, PSC demonstrated a main effect on psychological distress reduction [41], consistent with previous research showing PSC to be a predictor of employees’ psychological health [42]. In contrast to earlier findings [21], there was no evidence that client satisfaction was related to the effectiveness of EAPs. Moreover, satisfaction with an EAP did not have a moderating effect on changes in psychological distress as predicted. The change in psychological distress was not dependent on the level of satisfaction with the service.

### 4.1. Theoretical Implications

There are three main theoretical implications of the study. First, our study is one of the first to contend that attempts to modify worker psychological health through individually focused attempts (i.e., through EAPs) are likely to be effective conditional on the workplace climate. The findings support the role of Psychosocial Safety Climate in theory in this regard, specifically the secondary role PSC plays in conditioning the impact of other factors that affect psychological wellbeing [41]. The findings of this investigation complement those of earlier studies [41,42,43,44]. Individual interventions such as EAPs may significantly improve psychological health as shown here, but the strength of this effect is dependent on the corporate climate. More broadly, these findings have significant implications for any individual-level focused interventions. In general, research indicates that individual interventions are effective, but require upstream factors such as corporate climate to bolster long-lasting impacts. Our research suggests that the corporate climate in combination with individual focused interventions like EAPs should produce better predictive outcomes. These findings accord with previous research that found that naturally occurring mindfulness in employees positively affects creativity, skill discretion, psychological health and wellbeing. However, only in the context of high PSC levels were those mindfulness interventions sustainable [68].

The second theoretical implication relates to the origins of worker distress. This study supports key tenets of PSC theory [37]. Firstly, that PSC is a preeminent cause of distress, and second, that workers’ psychological health is fundamentally linked to the organisational climate. Taken together, this implies that workers’ psychological distress has two component processes, one that emanates from personal processes and is amenable to individually focused intervention, and another that emanates from the corporate work context. Our results show that while PSC improved, it did not change significantly as a result of the EAP. The observation that distress changed but not PSC indicates that EAPs are likely successful at reducing thoughts and feelings of distress but not helpful in changing the workplace climate. Hence EAPs appear to have a tertiary prevention function (reducing symptoms) rather than a primary prevention function (tackling problems at their source).

The study’s findings clearly inform the practical theory about the value and efficacy of EAPs and the evaluation of such programs e.g., [12,26]. This is the first study to analyse the relationship between an organisation’s corporate climate (PSC) and resulting psychological distress following an EAP intervention. This significantly extends the existing body of knowledge and application of PSC and has great theoretical implications for future research.

Furthermore, this study has been able to draw relationships between EAPs and psychological health and wellbeing in workplaces. The extension of research has increased the awareness of the role of corporate climate on employee psychological health and wellbeing, as well as the proposed use of the PSC as a reliable predictor of employee psychological health outcomes following EAP interventions. This study indicates PSC can have greater improvements on employee psychological distress than solely received from an EAP. Therefore, the theory that EAPs can improve psychological health is supported, however the depth or extent of benefits remains unknown. When implemented as a sole initiative, an EAP alone may not have the intended effect. The expected finding that PSC scores did not change over time can be explained by the focus of EAPs, which target individual improvement only rather than creating lasting change at an organisational level.

The lack of significant findings to support a correlation between client satisfaction and psychological health and wellbeing suggests that satisfaction surveys may not be an adequate or appropriate measure of EAP effectiveness within an organisation.

Findings from this study were also found to be consistent with PSC theory behind the Australian and the New Zealand Workplace Barometers [41,69], and evidence whereby PSC is a strong predictor of psychosocial health.

### 4.2. Practical Implications

According to the GHQ-12 cut-off scores (11/12) for being at risk of psychological disorder diagnosis, the sample at both Time 1 had a mean score well above 12, indicating that the sample required effective psychological interventions. However, a major implication from this research is that without giving attention to the workplace contexts from which clients emerge, the effectiveness of EAP interventions has constraints. Our results confirmed the important role of EAPs in reducing psychological distress and showed how the PSC of workplaces can directly influence employees’ psychological distress and moderate the impact of the EAP. This highlights the potential need for a shift in focus from an individual level to a focus on helping employers improve PSC at an organisational level. As PSC predicts psychological distress, businesses can use the PSC framework to improve working conditions and in return, workers health. Although the link between EAPs and HRM have been noted previously, our research really underscores the fundamental expanded role for EAPs in the future to insist on corporate evidence of the PSC or to play a role in assessing it, in the interests of providing an effective service.

Evaluating the effectiveness of EAPs has widespread benefits for employers seeking to reduce the costs of ineffective programs (i.e., workers compensation claims, lawsuits, absenteeism, staff turnover). Ultimately, the goal of this study and the implications may create healthier work environments and employees. Practical implications of this research involve increasing awareness of EAPs and their effectiveness in improving employee psychological health and wellbeing. For employers and EAP providers alike, with evidence informed decisions can be made regarding the implementation of EAPs within their organizations, ultimately facilitating an increase in the effective delivery of these initiatives.

### 4.3. Limitations

The repeated measures evaluation was a strength in the study design, but there are some limitations. The short time frame for evaluation may have meant that there was not enough time for the intervention to have a stronger effect. Our time frame of around one month for follow up was made in consultation with the EAP association to enable time for proactively scheduling appointments and for participants to attend EAP sessions. Even so, we still found significant person level (cross-level) main and interaction effects for PSC, indicating that with a larger sample, the effects would be larger. Notwithstanding, the lack of main and interaction effects of satisfaction with service may have been affected by the sample size. In our sample, dropouts had no difference on most demographic variables, reducing bias in the sample due to missingness whilst potentially improving the generalizability of the results.

The dispersion of surveys was within a consistent time frame for each individual. There was reduction in the Time 2 sample perhaps because some participants did not end up attending EAP sessions. We relied on self-reported responses which could create possible bias and social desirability effects. We also did not have information on the precise kind of intervention and our results may be stronger or weaker depending on the intervention implemented. A solution to this limitation might be to incorporate more objective measures in future research, perhaps reported by the EAP, such as the type of assistance utilised. Additionally, there are known historical differences in the development of EAPs between countries. While the results are likely generalisable in Australia and New Zealand, replication in other countries with historical differences is required.

Finally, we proceeded with a mixed effects model even though the ICC-1 was not significant. Bliese and colleagues ([67], p. 3) provide evidence that there is no penalty for using mixed-effects in this scenario and that “failure to use mixed effects to model nested data can increase the risk of a type I error (for group-level effects) and type II error (for lower-level effects)”.

One issue with the EAP satisfaction moderation test is that since satisfaction was assessed following the EAP is that clients might assess the EAP less favourably because their psychological health was not improving. This issue does not apply to the PSC moderation because PSC was assessed prior to the EAP.

### 4.4. Future Research

This research responds to calls for more evaluation of the effectiveness of EAPs and their potential contribution to organisational and employee outcomes [21]. Further research could collect repeated measures data, increase the number of participations and analyse the longer-term effects of EAPs. This would enable participants to receive a greater number of sessions, enable more participants to be gathered, as well as potentially increase the return of survey data from Time 2. Opting for other measures or scales of psychological functioning or categorising types of psychological distress (e.g., distress because of personal or work life) may also enable findings to be generalised to other domains, thus unearthing expanded functions for EAPs. Using a control group would assist in ruling out competing explanations.

Future research may also benefit from incorporating cost-benefit analyses to link EAP effectiveness evaluations to workplace productivity to inform HRM of strategies other than cost-cutting to improve productivity. Lastly, the findings from this study suggest the need for increased evaluation of PSC because of its moderating effect on psychological distress. Targeting this influence in conjunction with EAPs will be highly beneficial in follow-up studies as we continue to consider how to further develop EAPs to create change at an organisational level.

## 5. Conclusions

Through evaluating the effectiveness of EAPs, we observed expected improvements in employee psychological health and wellbeing. Importantly, PSC was found to moderate the efficacy of EAPs; the most beneficial effects in reducing psychological distress were found when employees were in high PSC contexts. EAP satisfaction was not found to moderate the effect of EAPs on psychological distress. If EAPs focused on improving PSC then we should begin to see clear benefits of EAPs for employee psychological health and wellbeing via both individual and organisational pathways.

## Figures and Tables

**Figure 1 ijerph-19-05067-f001:**
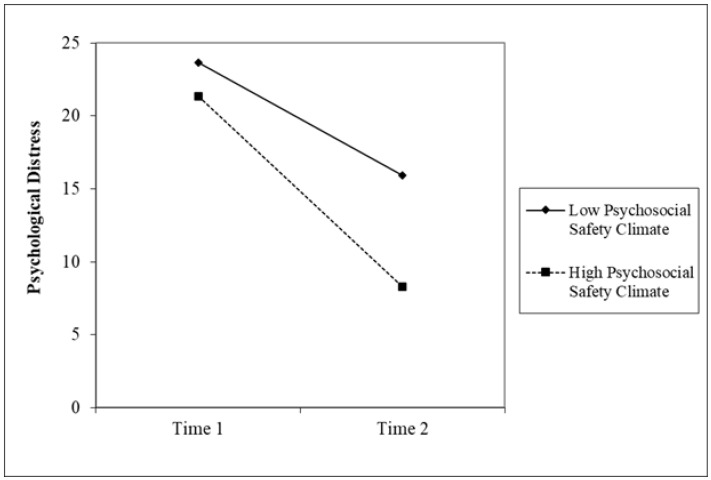
PSC × Time interaction to predict GHQ.

**Table 1 ijerph-19-05067-t001:** Descriptive Statistics; Means, Standard Deviations and Correlations.

	*M*	*SD*	*n*	1	2	3	4	5
1. EAP Completed	1.88	0.78	25	1				
2. GHQ (T1)	22.40	6.80	25	−0.08	1			
3. GHQ (T2)	12.21	7.22	25	0.45 *	0.02	1		
4. PSC (T1)	34.99	12.45	25	−0.39	−0.17	−0.53 **	1	
5. PSC (T2)	37.56	1.79	25	0.09	0.04	−0.14	0.22	1
6. Satisfaction (T2)	16.96	1.88	25	0.31	−0.03	−0.18	0.16	0.27

*Note.* * *p* < 0.05. ** *p* < 0.01. (two-tailed). Matched sample. EAP = Employee Assistance Program, GHQ = General Health Questionnaire, PSC = Psychosocial Safety Climate, T1 = Time 1, T2 = Time 2.

**Table 2 ijerph-19-05067-t002:** Final estimation of fixed effect; Predicting Change in Psychological Distress.

Fixed Effect	Coefficient	*SE*	*t*-Ratio	*df*	*p*
For Intercept 1					
Intercept	32.59	2.75	11.87	23	<0.001
PSC (T1)	0.12	0.18	0.67	23	0.51
For TIME slope					
Intercept 2	−10.19	1.66	−6.13	23	<0.001
PSC (T1)	−0.21	0.10	−2.17	23	0.041

*Note*. *SE* = Standard Error, *df* = Degrees of freedom, *p =* Probability, PSC = Psychosocial Safety climate, T1 = Time One.

**Table 3 ijerph-19-05067-t003:** Final estimation of fixed effects; Predicting Change in Satisfaction.

Fixed Effect	Coefficient	*SE*	*t*-Ratio	*df*	*p*
For Intercept 1					
Intercept 2	32.60	3.41	9.55	22	<0.001
Satisfaction (T2)	0.44	1.32	0.33	22	0.74
For TIME slope					
Intercept 2	−9.86	2.05	−4.81	22	<0.001
Satisfaction (T2)	−0.56	0.83	−0.67	22	0.507

*Note*. *SE* = Standard Error, *df* = Degrees of freedom, *p* = Probability, T2 = Time Two.

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
