# Peer review of "Contextualising the Effectiveness of an Employee Assistance Program Intervention on Psychological Health: The Role of Corporate Climate"

_ijerph, 2022, doi:10.3390/ijerph19095067_

Round 1
Reviewer 1 Report
The manuscript," Contextualising the Effectiveness of Employee Assistance Program Intervention on Psychological Health: The Role of Corporate Climate" presents an interesting approach to the collection information of the Psychological Health and Wellbeing of the employees. The “Introduction” section of the manuscript provide extensive revision and with a very good redaction. The review of literature is relevant to their study.
The aim of the study is properly highlighted and justified.
The manuscript, using cuantitative techniques based in pre/post-test design. The presentation of the technique and characterization of the results achieved indicate that the method is quite suitable and in fact could be useful to profundize this aspects in the the health sciences. However, I would like to receive more information about the data collection and procedure (pag.7):
- Were the participants paid?
- Did the project pass through an ethics committee?
- Could you give an example of an EAP session??
- Was it a convenience sample or what type of sample?
Overall the results are compelling and indicate that the method is more than suitable. Rigorously, the analisys are detailed. The authors do a very good job of presenting a methodology of the accuracy and precision of their results and demonstrate the suitability of the method. Further, the manuscript presents a good and actualized bibliography. The study is of interest for the scientific community.
Author Response
I would like to appreciate your consideration and constructive comments on our paper.
Reply to the reviewer Comments and Suggestions for Authors:
1. Were the participants paid?
R: ‘voluntarily’ participation has been mentioned
2. Did the project pass through an ethics committee?
R: ‘ethics protocol number: 202087’ has been mentioned
3. Could you give an example of an EAP session??
R: A typical EAP session involved confidential counselling, usually short-term and solution focussed, to support worker wellbeing.
4. Was it a convenience sample or what type of sample?
R: ‘convenience’ sample is now mentioned
Reviewer 2 Report
This manuscript, titled “Contextualizing the Effectiveness of Employee Assistance Program Intervention on Psychological Health: The Role of Corporate Climate,” examined the moderating effect of psychosocial safety climate (PSC) on the effectiveness of employee assistance programs (EAPs) on mental health. The authors are commended for a well-written paper with an adequate literature review of previous studies, the logical rationale for the research question, and sophisticated statistical analyses.
There are several issues that need to be addressed.
- Since PSC is a main theme in the study, it would be helpful to provide a detailed definition and one or two examples. Although such information can be found in section 1.3, it would provide readers with a clear context if a clear definition is provided in the overview. The PSC theory (line 36) also needs more elaboration.
- Section 1.2 can be trimmed down, and Section 1.4 needs more elaboration. In particular, Section 1.4 should discuss in relation to the moderating effect of client satisfaction.
- Were the EAPs of these companies similar? Were the types of assistance utilized different among participants? How would these affect the interpretation of the findings?
- It is not clear how many participants were recruited. Lines 226-228 indicate 25 participants, but lines 319-320 indicate 28-128.
- Line 237: “The independent variable was the EAP intervention (individual sessions) delivered” – The IV of the study was time (before, after). Not EAP.
- Lines 348-349: “Seniority (i.e., employee, middle manager, manager) was found to be moderately negatively correlated with gender, r = -.49, n = 25, p < .05.” Since both seniority and gender were categorical variables, should use a Chi-square instead.
- Table 2: Some of the analyses should be performed using ANOVA since the variables involve more than 2 categories.
- Lines 381-383: Simple slope analysis results should be reported as well. Specifically, it is important to know whether the simple slopes for high PSC and low PSC were significant. By looking at the gradient of the slopes, it looks like both slopes were significant. Whether both slopes were significant or only one was significant, it could be an interesting point of discussion in Section 4 by relating the findings with those from previous studies.
- Section 4.1 has no citations.
- Section 4.2 can be enriched by focusing on PSC.
- Kindly proofread the manuscript throughout for minor mistakes, e.g.,
- Line 37: “… plays a significant role to play in determining…”
- Line 123: “… not only for enriching workers psychological health…”
- Line 175: “Employee Assistance Program” should be "EAP."
Author Response
R: The Abstract has been updated to include a brief definition of PSC as below
- ‘psychosocial safety climate (PSC) is the corporate climate for workers' psychological health and safety,
- Psychosocial safety climate (PSC) refers to the organisational climate for employee psychological health and safety, and encompasses the way management value and prioritise worker psychological health. PSC theory
- We also added the section below
- As a facet of workplace climate, PSC refers to employees’ perception of managerial policies, practices, and procedures implemented in their organization to protect workers' psychological health and safety and prevent psychological injury due to workplace psychosocial risks. Indeed, PSC protects employees’ psychological health and safety through four principles: management commitment, management priorities, organizational communications and participation [22].
- Section 1.2 can be trimmed down, and Section 1.4 needs more elaboration. In particular, Section 1.4 should discuss in relation to the moderating effect of client satisfaction.
R: Section 1.2 has been trimmed down. Section 1.4 explains the moderation proposition.
- Were the EAPs of these companies similar? Were the types of assistance utilized different among participants? How would these affect the interpretation of the findings?
R: Also we did not have information on the precise kind of intervention and our results may be stronger or weaker depending on the intervention. A solution to this limitation might be to incorporate more objective measures in future research, perhaps reported by the EAP, such as the type of assistance utilised.
- It is not clear how many participants were recruited. Lines 226-228 indicate 25 participants, but lines 319-320 indicate 28-128.
- The statement was slightly incorrect. Now modified.
- R: Data were collected from Australian and New Zealand participants at pre- (n = 128) and post- (n = 28) EAP completion. At Time 2, 25 cases were successfully matched with data from Time 1 using participant email addresses. Data were cleaned, coded, screened, and analysed using the SPSS-V26 software.
- Line 237: “The independent variable was the EAP intervention (individual sessions) delivered” – The IV of the study was time (before, after). Not EAP.
R: Reworked as ‘The independent variable was time, which reflects the EAP intervention (individual sessions) delivered within an approximate one-month time frame.’
- Lines 348-349: “Seniority (i.e., employee, middle manager, manager) was found to be moderately negatively correlated with gender, r = -.49, n = 25, p < .05.” Since both seniority and gender were categorical variables, should use a Chi-square instead.
R: That’s correct! Thank! Indeed, the correlation between seniority and gender is not presented in Table 1. We conducted a Chi-square test for seniority and gender and reported the results on page 10. Therefore, we removed the sentence “Seniority (i.e., employee, middle manager, manager) was found to be moderately negatively correlated with gender, r = -.49, n = 25, p < .05.” and updated it with Chi-square results.
Table 2: Some of the analyses should be performed using ANOVA since the variables involve more than 2 categories.
R: Chi-Square test was conducted, and the results are reported for age, gender, employment status, living location and seniority in text.
The Chi-Square test results reported in the text and table 2 is removed.
- Lines 381-383: Simple slope analysis results should be reported as well. Specifically, it is important to know whether the simple slopes for high PSC and low PSC were significant. By looking at the gradient of the slopes, it looks like both slopes were significant. Whether both slopes were significant or only one was significant, it could be an interesting point of discussion in Section 4 by relating the findings with those from previous studies.
R: To this end, we tested for the “Simple Intercepts and Slopes at Conditional Values” of PSC and reported the results as below.
Testing simple intercepts and slopes at conditional values of PSC indicated that both slopes were significant in high and low PSC situations. However, in high PSC (B = -12.87, t = 6.74, p <.001), a stronger significant slope was found compared with low PSC (B = -7.51, t = 8.59, p <.001).
- Section 4.1 has no citations.
- R: more citations were added in this section. Section 4.2 can be enriched by focusing on PSC.
R: yes, please see the section updated.
- Kindly proofread the manuscript throughout for minor mistakes, e.g.,
- Line 37: “… plays a significant role to play in determining…”
R: It has been fixed as “… plays a significant role in determining…”
- Line 123: “… not only for enriching workers psychological health…”
R: Fixed as “… not only for improving workers’ psychological health…”
- Line 175: “Employee Assistance Program” should be "EAP."
R: Fixed! “Employee Assistance Program” words were replaced with EAP for all the cases (except the first one).
Round 2
Reviewer 2 Report
Substantial changes were made to the manuscript. I would like to commend the authors for the effort put forth in making great improvement to the manuscript. There are some minor amendments needed to further improve the content.
- Section 1.4 still needs some improvement to provide the rationale for the hypothesized moderating effect of client satisfaction.
- Line 462-464 “However, in high PSC (B = -12.87, SE = 1.91, t =- 74, p <.001), a stronger significant slope was found compared with low PSC (B = -7.51, SE = 2.21, t = -3.40, p <.001).” This statement is valid only if an additional analysis is performed to compare the two regression slopes.
Author Response
Dear reviewer,
We took the opportunity to give the paper thorough editing in the review process. Please find my responses to your constructive comments below. Deeply grateful for all your consideration.
1. Section 1.4 still needs some improvement to provide the rationale for the hypothesized moderating effect of client satisfaction.
R: I'm not sure we need to justify the moderation fit in this case. However, I amended section 1.4 (p.5, lines: 209-216) as below. If you are happy with these updates, we need to add Schoonhoven (1981) to the references and update all the reference numbers.
Although others have examined the moderating effects of PSC, no one has yet investigated the moderating role of client satisfaction for future improvements through the lens of a pre-and post-test design. As mentioned by Schoonhoven, ‘When contingency theorists assert that there is a relationship between two variables ... which predicts a third variable, ... they are stating that an interaction exists between the first two variables’ (1981, p. 351). Therefore, the effectiveness of EAPs in relation to GHQ scores across time (T1 and T2), explains the rationale for hypothesizing a moderating effect.
2. Line 462-464 “However, in high PSC (B = -12.87, SE = 1.91, t =- 74, p <.001), a stronger significant slope was found compared with low PSC (B = -7.51, SE = 2.21, t = -3.40, p <.001).” This statement is valid only if an additional analysis is performed to compare the two regression slopes.
R: the additional test has been already applied. This section has been updated as below (p. 9, lines 390-394):
To compare those two regression slopes, an additional analysis was performed to test simple intercepts and slopes at conditional values of PSC indicated that both slopes were significant in high and low PSC situations. However, in high PSC (B = -12.87, SE = 1.91, t =- 6.74, p <.001), a stronger significant slope was found compared with low PSC (B = -7.51, SE = 2.21, t = -3.40, p <.001).
Should you have any further constructive comments or considerations, please do not hesitate to let me know.
Kind regards,
Ali